# A Seismic Data Acquisition System Based on Wireless Network Transmission

**DOI:** 10.3390/s21134308

**Published:** 2021-06-24

**Authors:** Yanxia Huang, Junlei Song, Wenqin Mo, Kaifeng Dong, Xiangning Wu, Jianyi Peng, Fang Jin

**Affiliations:** 1School of Computer Science, China University of Geosciences, Wuhan 430074, China; huangyanxia@cug.edu.cn (Y.H.); wxning@cug.edu.cn (X.W.); jianyi_peng@cug.edu.cn (J.P.); 2School of Automation, China University of Geosciences, Wuhan 430074, China; moon_qin@cug.edu.cn (W.M.); dongkf@cug.edu.cn (K.D.); jinfang78@cug.edu.cn (F.J.); 3Hubei Key Laboratory of Advanced Control and Intelligent Automation for Complex Systems, Wuhan 430074, China; 4Engineering Research Center of Intelligent Technology for Geo-Exploration, Ministry of Education, Wuhan 430074, China

**Keywords:** PMN-PT, PZT, piezoelectric accelerometer, seismic acquisition, FPGA

## Abstract

A seismic data acquisition system based on wireless network transmission is designed to improve the low-frequency response and low sensitivity of the existing acquisition system. The system comprises of a piezoelectric transducer, a high-resolution data acquisition system, and a wireless communication module. A seismic piezoelectric transducer based on a piezoelectric simply supported beam using PMN-PT is proposed. High sensitivity is obtained by using a new piezoelectric material PMN-PT, and a simply supported beam matching with the PMN-PT wafer is designed, which can provide a good low-frequency response. The data acquisition system includes an electronic circuit for charge conversion, filtering, and amplification, an FPGA, and a 24-bit analog-to-digital converter (ADC). The wireless communication was based on the ZigBee modules and the WiFi modules. The experimental results show that the application of the piezoelectric simply supported beam based on PMN-PT can effectively improve the sensitivity of the piezoelectric accelerometer by more than 190%, compared with the traditional PZT material. At low frequencies, the fidelity of the PMN-PT piezoelectric simply supported beam is better than that of a traditional central compressed model, which is an effective expansion of the bandwidth to the low-frequency region. The charge conversion, filtering, amplification, and digitization of the output signal of the piezoelectric transducer are processed and, finally, are wirelessly transmitted to the monitoring centre, achieving the design of a seismic data acquisition system based on wireless transmission.

## 1. Introduction

Seismic exploration technology is the most reliable and commonly used technology in oil exploration and plays an important role in the detection of oil reserves [1]. Traditional wired seismic exploration equipment has bulky cables making it very difficult to artificially deploy geophones in complex environments, thus making it impossible to meet the requirements of seismic exploration [2,3]. The rapid development of exploration equipment has mainly manifested in the following two aspects: the demand for mineral resources, which is continuously increasing, and seismic exploration areas, which have gradually developed into complex environments such as hills and rainforests [4,5]. The demand for “high-efficiency, wideband and high sensitivity“ equipment has prompted the development of seismic exploration instruments towards flexible and portable large-scale wireless ad hoc networks [6]. Additionally, the emergence of new technologies such as “big data” and the “internet of things” has further improved the performance of wireless communication local area networks, which has greatly helped the development of wireless seismic data acquisition systems in the field environment [7].

In seismic exploration, a general wireless sensor network cannot meet the needs of seismic exploration wireless acquisition due to the wider exploration area, the larger number of seismic accelerometers, and the huge amount of seismic data. Aiming to improve on the shortcomings of traditional seismic exploration data acquisition systems, a digital seismic data acquisition system, based on ZigBee and WiFi, is designed, as shown in Figure 1. The ZigBee protocol, with a slower signal transmission speed and lower computing capacity, is usually used at the end of the seismic acquisition network, but the cost and power consumption of using it are low [8]. In the core network of the wireless gateway, WiFi protocol, with high speed and real-time performance, is used to meet the needs of a seismic exploration system for power consumption and transmission rate [9]. A digital seismic data acquisition system is designed to meet the exploration requirements of high sensitivity and wideband, including the seismic accelerometers node, an analog-to-digital converter (ADC), and digital data acquisition module based on FPGA.

As a key component of seismic data acquisition, the development of seismic accelerometers directly influences the accuracy of exploration [10].

A seismic piezoelectric accelerometer is a vibration detector that converts vibration acceleration signals into electrical signals [11], and is composed of a piezoelectric (PE) transducer, integral electronics, a shell, and coupling elements. The PE transducer uses piezoelectric crystals, artificial polarized ceramics, or organic polymer piezoelectric materials as sensing elements [10,12]. There are three main transducer construction designs used in piezoelectric accelerometers: compression mode, shear mode, and bending or flexural mode [13,14]. A commonly used seismic piezoelectric accelerometer is a compressed detector based on thickness deformation, which has a simple structure and good performance with a high frequency bandwidth and high signal-to-noise ratio [13,14,15]. However, since a large mass is included inside the accelerometer and it has a poor low-frequency response, it fails to meet the standards of high-precision seismic exploration.

The sensitivity of piezoelectric accelerometers is greatly affected by the dielectric constant, piezoelectric constant, and electromechanical coupling coefficient of piezoelectric materials. In previous studies, lead zirconate titanate (PZT) ceramics were widely applied as sensing elements in piezoelectric accelerometers [15,16,17]. Due to the limited performance of the piezoelectric elements, the sensitivity of existing piezoelectric accelerometers is not high enough and needs to be improved.

As new types of piezoelectric materials, xPb(Mg1/3Nb2/3)O3-(1-x)PbTiO3 (PMN-PT or PMNT) relaxation ferroelectric single crystals with high piezoelectric performance have attracted continuous attention, the piezoelectric strain constant (d33) and electromechanical coupling coefficient (k33) could reach as high as 2000 pC/N and 0.94, respectively [18,19], and the comprehensive performance of PMN-PT is more superior than PZT [20]. The outstanding performance of PMN-PT makes it an ideal functional material for high sensitivity and low noise sensors. PMN-PT has been preliminarily used in ultrasonic motors [21], piezoelectric transformers [22], magnetoelectric sensors [23], and ultrasonic transducers [24]. These devices have a superior performance to those made with PZT, which shows promising applications in device fabrications. The sensitivity of the accelerometer will benefit from PMN-PT if it can be used as the sensing element of a seismic piezoelectric accelerometer.

However, high-precision seismic exploration is still limited by the poor low-frequency response of the traditional central compressed mode. The accelerometer in the bending mode has a good low-frequency response [14]. The commonly used bending mode accelerometer generally adopts a cantilever beam [25,26,27,28], which has high sensitivity, but with a larger size and a relatively narrow frequency band [27,28]. Levinzon designed an ultra-low-noise seismic piezoelectric accelerometer with a double cantilever beam [10], which is a cylindrical shape with a diameter of about 65 mm and a resonance frequency of about 370 Hz. The seismic piezoelectric accelerometer is linear, and the sensitivity is limited by the onset of nonlinearity [29]. The resonant accelerometers can tune the electrostatic nonlinearities to enhance the sensitivity [30], which has the advantages of being small in size with high precision [31,32]. Whereas, the resonant accelerometers are not suitable for measuring weak vibration and have a narrow bandwidth [31]. In addition, the structure, manufacturing and packaging processes of resonant accelerometers are complicated [33]. Seismic piezoelectric accelerometers are still the most commonly land-used sensor for acquiring seismic data. In summary, how to realize a piezoelectric accelerometer with high sensitivity, a good low-frequency response, and small size is a key problem to be solved in the exploration domain.

To solve the aforementioned problems, this paper achieves a PE transducer with a simply supported beam structure, using PMN-PT as its piezoelectric element, which has the advantages of having high sensitivity, a good low-frequency response performance, and being small in size. The charge conversion, filtering, amplification, and digitization of the output signal of the PE transducer are processed and are finally wirelessly transmitted to the monitoring centre, realizing the design of a seismic data acquisition system based on wireless transmission.

This paper is organized as follows. Firstly, the piezoelectric transducer of simple beam type is designed and verified by simulation, and then the charge conversion, filtering, and amplification, digital processing is designed. Finally, the experimental verification is carried out.

## 2. Design and Implementation

The system mainly includes a piezoelectric transducer, charge conversion circuit, high and low pass filter, amplifier, the ADC, and the microcontroller FPGA, as shown in Figure 2. The piezoelectric transducer converts the vibration signal into an electrical signal and outputs it in the form of an electric charge. The charge conversion circuit converts the high-impedance charge signal into a low-impedance voltage signal and sends it to a high and low pass filter for filtering then amplifies it by an amplifier circuit, and finally uses an ADC to digitize the acceleration signal and send it to the FPGA. These whole procedures achieve the functions of data acquisition and data transmission of the seismic data acquisition system. The wireless transmission network sends the seismic data sampled by FPGA to the monitoring centre to analyse and process the signal for real-time monitoring and measurement.

### 2.1. Piezoelectric Transducer-Capturing Raw Seismic Data

The structure of the piezoelectric accelerometer is shown in Figure 3a. The PE transducer and the charge amplifier circuit are all embedded in the accelerometer. As shown in Figure 3b, the PE transducer proposed in the paper consists of three parts: a simply supported beam, piezoelectric element, and mass. The PE transducer vibrates with the vibration generated by the artificial excitation source, causing deformation of the piezoelectric element and the supported beam. The deformation of the piezoelectric element will transform the mechanical energy of the vibration into an electrical signal, which is known as the positive piezoelectric effect. The spring is a caging device to enhance the shock resistance of the piezoelectric accelerometer.

A simplified simulation model of the PE transducer with the simply supported beam is established in a multi-physical field analysis software COMSOL 5.2. As shown in Figure 4, the size of the simply supported beam is 25 mm × 10 mm × 0.6 mm (length × width × thickness), and the Mass is 10 mm × 10 mm × 5 mm (length × width × thickness). The piezoelectric element and mass in the model are closely combined and located in the middle of the simply supported beam. The piezoelectric element has a polarization direction of [001] and a size of 10 mm × 10 mm × 1 mm. We use PZT and PMN-PT as piezoelectric elements for simulation experiments and compare their performance.

In the simulation, the analysis of the electromechanical coupling relationship is mainly carried out, and the physical fields involved are the circuit, the solid, and the electrostatic field. There are three main simulation steps: modelling, loading solution, and result analysis. First, establish the designed model, define the required parameters and functions, add the required physical fields, perform meshing, then define the solving equation, solve the required physical quantities, and finally analyse the results. The damping loss factor is the main method to describe spillage of material in the frequency domain and the damping type in the simulation is the isotropic loss factor, which can be user-defined. The boundary condition of the simply supported beam is set as zero displacements and bending moment at both ends of the simply supported beam:(1)$$\begin{array}{l} Y_{x=0}=0,{\frac{d^{2}Y}{dx^{2}}|}_{x=0}=0\\ Y_{x=l}=0,{\frac{d^{2}Y}{dx^{2}}|}_{x=l}=0 \end{array}$$
where *l* is the length of the simply supported beam, and the solver is a modal solver.

#### 2.1.1. Force Simulation in Different Directions

Seismic waves are transmitted to the accelerometer from the points of explosion during seismic prospecting. The vibration directions of the seismic waves are random which can be considered into *X*, *Y*, and *Z* directions. The vibration in one special direction, such as the *Z* direction, needs to be detected accurately by a one-component accelerometer. At the same time, the accelerometer should have the ability to suppress vibration interference in other directions, which is usually evaluated using transverse sensitivity. In this part, the force applied to the model is simulated along *X*, *Y*, and *Z* directions, respectively, the vibration acceleration is always 3 m/s^2^, and PMN-PT is used as a piezoelectric element.

As shown in Figure 5, the simulation model is very sensitive to the vibration in the direction of *Z*, but not sensitive to the vibration of *X* and *Y* directions. The output voltage of the model in different directions is shown in Figure 5. In the 0–1000 Hz band, the output voltage of the model in the *Z* direction is 85–101.6 mV, and the voltage variation is 16.6 mV. Whereas the output voltages of the model along *X* and *Y* directions are 0.049 mV and 0.48 mV, respectively, which are far less than the output voltage in the *Z* direction. The transverse sensitivity is less than 0.56%, indicating that the model can effectively suppress transverse interference. In addition, when the simulation model is subjected to vibrations with the same acceleration and different frequencies, the output voltage is different. The higher the vibration frequency usually combines the higher the output voltage. In summary, the piezoelectric simply supported beam model employed in this paper is very sensitive to vibration acceleration and has great transverse interference suppression, which is suitable for designing the seismic piezoelectric accelerometer.

#### 2.1.2. Different Piezoelectric Elements

In the simulation experiment of different piezoelectric elements, PZT-5A is used as the piezoelectric element in comparison with the same size of relaxation ferroelectric single crystal PMN-PT. The output voltage is related to the parameters of the piezoelectric element. The comparison table for the material properties of PMN-PT and PZT is shown in Table 1. The impedance spectrum of the different elements is shown in Figure 6a. It can be seen that the impedance of the PZT element is significantly higher than that of PMN-PT. Consider the piezoelectric element as a voltage source, when the output impedance is constant, the larger the voltage source impedance combines the smaller the output voltage. Acceleration-invariant and frequency-varying vibrations are applied to a length of 25 mm simply supported beam along the *Z* direction, and the output voltages of the models based on PZT-5A and PMN-PT at different frequencies are shown in Figure 6b. The resonant frequency of the piezoelectric simply supported beam based on PMN-PT is 2500 Hz, while the PZT is 2450 Hz, which is higher than the maximum frequency of a seismic wave signal with a bandwidth of 0–800 Hz. Therefore, the frequency bandwidth of the piezoelectric simply supported beam can meet the design requirement of seismic accelerometers. It can also be seen from Figure 6b that the response of the beam based on PMN-PT to vibration is superior to that of the beam based on PZT. The output voltage of the beam based on PMN-PT is 95% higher than that based on PZT in 0–1000 Hz.

For a seismic PE transducer, the sensitivity is the ratio of the transducer’s output voltage variation to the vibration acceleration variation [13]:(2)$$S_{V}=\frac{V_{OUT}}{a}$$

In the simulation experiment, we measured the voltage sensitivity. In general, the sensitivities are divided into charge sensitivity and voltage sensitivity [13], $S_{Q}=S_{V}*c$, where *c* is the sum of the capacitance of the piezoelectric element and the attached cable.

The sensitivity of the models based on PMN-PT and PZT are 28.38–52.03 mV/ms^−2^ and 14.5–23.08 mV/ms^−2^ in 0–1500 Hz, respectively, shown in Figure 7. The sensitivity of the model based on PMN-PT is 95% higher than that of the model based on PZT, indicating that PMN-PT can significantly improve the sensitivity of the PE transducer model.

#### 2.1.3. Different Structural Models

Central compressed structure based on thickness deformation has been widely used in PE transducers. In this simulation, a central compressed model (schematic in Figure 8) is compared with the simply supported beam model. The piezoelectric element of the models is PMN-PT of 10 mm × 10 mm × 1 mm and the Mass is 10 mm × 10 mm × 5 mm.

Figure 9 shows that the sensitivity of the simply supported beam model and the central compressed model is 28.38–36.61 mV/ms^−2^ and 10.56–10.57 mV/ms^−2^ in 0–1000 Hz, respectively. It can be seen from the sensitivity of the simply supported beam for the same vibration is twice more than that of the central compressed structure. The construction of the simply supported beam provides mechanical amplification of motion giving the high values of the sensitivity of the PE transducer. Due to the high resonance frequency and low sensitivity of the central compressed model, its sensitivity is relatively stable within 1–1000 Hz, while the sensitivity of the simply supported beam increases as the frequency. Seismic wave signals have a loss in the process of propagation, and the higher the frequency of the seismic wave combines the greater the amplitude attenuation [34]. The sensitivity of the piezoelectric simply supported beam model increases with the increase of frequency under a certain acceleration, which can compensate for the attenuation of the seismic wave amplitude with frequency to a certain extent.

The above multiple simulation results show that the piezoelectric simply supported beam can be used to improve the sensitivity of a seismic PE transducer. The response of the piezoelectric simply supported beam with PMN-PT to vibration is better than the PZT-5A, which is 95% higher than that of the PZT model. And the sensitivity of the simply supported beam at the same vibration is twice larger than that of the commonly used central compressed structure.

### 2.2. The Analog Circuit–Charge Conversion, Filtering, and Amplification

#### 2.2.1. Charge Conversion Circuit

The charge conversion circuit (Figure 10) converts the high-impedance charge signal output by the piezoelectric transducer into a low-impedance voltage signal. The advantage is that the output voltage is only related to the magnitude of the charge Q and the value of the feedback capacitor *C_f_*, the output voltage:(3)$$U_{o}=\frac{Q}{C_{f}}$$

The operational amplifier uses TL084, the feedback capacitor *C_f_* uses CB10 polystyrene film capacitor with *C_f_* = 1000 pF, and feedback resistance *R_f_* = 1 GΩ. The adjustable DC balancing resistor *R*_3_ is to reduce the offset current and offset voltage. The purpose of parallel capacitor *C*_2_ at both ends of *R*_1_ is to realize phase compensation and to avoid the self-excited oscillation caused by the pole formed of the *R*_1_ and the input capacitor of the op-amp, and *R*_1_
*≈ R*_3_. *C*_1_ is a DC blocking capacitor, which is used to isolate the DC offset of the seismic accelerometer; *R*_2_, *R*_4_, and *R*_5_ form a T-type resistor network, which is equivalent to obtain a feedback resistance of 1 GΩ at both ends of *R*_2_ and *R*_5_.

#### 2.2.2. Filter and Amplifier Circuit

The piezoelectric accelerometer is a weakly damped vibration system, so its amplitude-frequency characteristic has a strong resonance peak in the high-frequency band, which introduces a lot of high-frequency noise, causing great interference or even distortion to the signal. Therefore, this paper designs a Butterworth second-order low-pass filter to extract useful low-frequency effective signals. According to the bandwidth of a PE transducer, the cut-off frequency is set as 1.5 kHz and the amplification gain is 2. Based on the characteristics of the Butterworth filter, the quality factor is set as 0.707, the capacitance *C*_9_ = *C*_10_ is 10 nF, the low-pass cut-off frequency of the circuit is dependent on its RC time constant according to the following relationship:(4)$$f_{c}=\frac{1}{2\pi \sqrt{R_{6}R_{7}C_{9}C_{10}}}=1500\;\mathrm{Hz}$$

*R*_6_ = *R*_7_ is calculated as 10 kΩ; Set *R*_8_ = *R*_9_ as 10 kΩ and the gain is 2, as shown in Figure 11.

Due to various reasons, the circuit is prone to zero drift when the charge is converted into a voltage. Thus, this paper designs a high-pass filter after the low-pass filter circuit to filter out the unnecessary DC part of the circuit. In addition, the filter circuit must also have the function of output amplification. After the signal is amplified to a reasonable range, data acquisition is performed. The input impedance of the non-inverting amplifier is high, but the output impedance is low. It has relatively little effect on the output signal. Hence, the non-inverting proportional amplifier is used in the paper, and the circuit schematic diagram is shown in Figure 11.

Set the cut-off frequency of the high-pass filter as 1 Hz to collect more low-frequency signals, then calculated the results as *C*_11_ = *C*_12_ = 4.7 μF, and *R*_11_ = 10 kΩ, *R*_12_ = 150 kΩ. The op-amp TL084 and resistors *R*_10_ and *R*_13_ form the non-inverting amplifier circuit. The output gain of the entire circuit can be changed by changing the gain of the non-inverting amplifier circuit, and the output signal can be guaranteed to maintain in-phase with the piezoelectric transducer. For piezoelectric transducers with different sensitivities, a suitable magnification should be selected to adapt the output range. Thus, *R*_13_ is set as an adjustable resistor, and the magnification can be adjusted according to actual needs. According to the equation of gain (Equation (5)), the corresponding resistance values of *R*_13_ and *R*_12_ can be calculated.
(5)$$Av=1+\frac{R_{13}}{R_{12}}$$

The overall gain of the circuit is mainly achieved through the gain of this stage (i.e., U1C TL084D), and this gain needs to be designed based on two factors: the maximum expected voltage that the PE transducers generated and the maximum voltage that the ADC support (5 V in our case). For example, when the gain is set to 25, then *R*_12_ = 10 kΩ and *R*_13_ = 240 kΩ. Then the PCB board is made for a smaller seismic accelerometer.

### 2.3. The FPGA and ADC—Sampling and Digitization

After the raw seismic signal is converted, filtered, amplified, and further sampled and quantized with a resolution of 24 bits/sample for digital transmission.

The voltage signal output by the amplifying circuit is converted from analog to digital by the ADC ADS1256 and sent to FPGA for data processing and transmission, including a process of saving the data to an SD card and sending the data to ZigBee to be wirelessly transmitted to the gateway in real-time.

The ADS1256 is a greatly low-noise, 24-bit analog-to-digital (A/D) converter, which can provide a data rate of up to 30 k samples per second (SPS) and an internal SPI serial interface to connect with FPGA.

The FPGA selected in this paper is the Cyclone IV series chip EP4CE6F17C8 of Altera Company, which provides 68 IO ports, 5 V power supply, 3.3 V power supply, multiple GND, 40-pin expansion port, and other resources.

The use of Verilog HDL programming to control ADS1256 can realize the sampling, processing, storage, and serial port transmission of seismic data. The flow chart of data processing is shown in Figure 12.

For the visual display of the data and the convenience of subsequent processing, this paper converts the collected 24-bit data into decimal and then converts it into ASCII code for output.

During the hexadecimal conversion, the positive and negative values of the collected data are firstly judged, and then the hexadecimal results are calculated according to the following formula.

If the output is positive:(6)$$\mathrm{Voltage}\;\mathrm{value}=\frac{2\times V_{REF}}{PGA\times (2^{23}-1)}\times (ADC\;output\;data)$$

Else the output is negative:(7)$$\mathrm{Voltage}\;\mathrm{value}=\frac{2\times V_{REF}}{PGA\times (2^{23}-1)}\times (0xFFFFFF-ADC\;output\;data+1)$$

The obtained hexadecimal results are decoded and converted into BCD code. Then the BCD code plus 48 to convert into an ASCII value and sent to the Zigbee module through the serial port.

## 3. Results

### 3.1. Test of Piezoelectric Transducer

In this experiment, we built a sensor calibration control system and developed PE transducers by using the simply supported beam with PMN-PT and PZT. Figure 13 shows the structure diagram of the control system, which consists of a calibration controller (WS-5931, Beijing Wavespectrum Science & Technology Co., Ltd., Beijing, China), a power amplifier (GF-500, Beijing Wavespectrum Science & Technology Co., Ltd., Beijing, China), a vibration exciter (JZ-50, Beijing Wavespectrum Science & Technology Co., Ltd., Beijing, China), a charge amplifier (WS-2401, Beijing Wavespectrum Science & Technology Co., Ltd., Beijing, China), a standard acceleration sensor (CK8305, Beijing Wavespectrum Science & Technology Co., Ltd., Beijing, China, which is a central compressed accelerometer with PZT as the piezoelectric element), the PE transducer developed in this paper, and PC. The PC controls the calibration controller to provide excitation signals for the vibration exciter, and the vibration exciter is driven at various voltages and frequencies by the power amplifier. At the same time, the output signals from the standard acceleration sensor and the PE transducer are imported to the calibration controller after amplifying by the charge amplifier, which is monitored by computer software.

The dimensions of the piezoelectric elements and the simply supported beam are shown in Table 2. The orientations of the piezoelectric elements are oriented along [110]. The two elements are both polarized along the thickness direction. Two electrodes are pasted on the two surfaces perpendicular to the thickness direction of each piezoelectric element. The piezoelectric element with the electrodes is pasted between the Mass and the simply supported beam.

Two central compressed transducers and two simply supported beam transducers based on PZT and PMN-PT are made for conducting their sensitivity tests. According to the rules of sensitivity testing in engineering practice, the sensitivity of the transducers was tested at 160 Hz, which is shown in Table 3.

The experimental result shows that a piezoelectric simply supported beam can be used to improve the sensitivity of the accelerometer. The sensitivity of the PMN-PT model to vibration is significantly excellent. The application of PMN-PT to the central compressed model and the piezoelectric simply supported beam can increase the sensitivity of the transducers by more than 130% and 190% compared to the traditional PZT, respectively. The sensitivity of the simply supported beam based on PMN-PT is twice higher than that of the commonly used central compressed structure based on PZT.

We obtained the frequency response curve of the simply supported beam transducer with a length of 25 mm based on PMN-PT and PZT. In Figure 14, it can be seen that the sensitivity increases with the increasing frequency and the maximum sensitivity is at 1050 Hz. In other words, the resonance frequency of the model is 1050 Hz, which is higher than the maximum frequency of a common seismic wave signal (0–800 Hz). In 0–1000 Hz, the sensitivity changing trend is consistent with the conclusion obtained by the simulation calculation in Section 2.1. The sensitivity of PMN-PT is significantly higher than that of PZT, indicating that the application of PMN-PT can significantly improve the model’s performance. Similarly, PMN-PT material is expected to increase the sensitivity of other piezoelectric sensors.

There is a certain error between the simulation results and the experimental results. This might be due to the fact in our experiments the connection type between mass, piezoelectric material, and the beam cannot be completely consistent with the simulation conditions, the certain errors from the control system (including the inherent error of the exciter and measurement system, the coupling error between exciter and PE transducer), and the interference introduced by the external environment, etc.

The sensitivity of the simply supported beam model and the central compressed model is shown in Figure 15. The sensitivity of the simply supported beam for the same vibration is twice higher than that of the commonly used central compressed structure, which is consistent with the simulation result. The construction of the simply supported beam provides mechanical amplification of motion, giving the high values of the sensitivity of the PE transducer.

To verify the low-frequency response characteristics of the simply supported beam transducer and the central compressed transducer, we provide them with 1 Hz sine wave vibration. Figure 16 shows the output of the transducers. It can be seen that the output amplitude of the simply supported beam transducer is large and the signal distortion is small, but the output signal of the central compressed transducer is significantly distorted. The fidelity of the simply supported beam transducer in low frequencies is better than that of the central compressed transducer. Therefore, using a simply supported beam transducer instead of a central compressed transducer has the potential to improve the low-frequency response characteristics of a seismic piezoelectric accelerometer. Bending beam structures, such as simply supported beams, can easily pick up vibration signals and can also be used in the design of resonant accelerometers.

### 3.2. Output Results of the Data Acquisition System

The model of the piezoelectric transducer is fixed on the vibration exciter, and the collected signals are output to the designed charge amplifier circuit. According to the results of the previous tests, the charge sensitivity of the piezoelectric transducer model is 2.48 pC/ms^−2^. Applying a sine wave signal with a frequency of 160 Hz and acceleration of 15.8 m/s^2^ on the vibrating exciter, the voltage value should be 39.18 mV by the calculation formula (3) of the charge conversion circuit. The magnification of the charge amplifier circuit is set to 50 times, the voltage sensitivity of the piezoelectric accelerometer after the charge amplifier is 124 mV/ms^−2^, and the theoretical output voltage value should be 1.959 V. The oscilloscope is used to observe and measure the output of the designed accelerometer with the charge amplifier circuit, as shown in Figure 17. The output voltage is 1.98 V, and the theoretical error is 1.07%. Given the system error, the error is within the normal range.

ADS1256 is controlled by FPGA to collect and digitize the output signal of the charge amplifier. The waveform of the signal collected by FPGA can be completely observed in the logic analyser SignalTap, as shown in Figure 18.

The serial port assistant can receive the data sent by FPGA and display it in real-time, indicating that the data can be sent to the ZigBee module through the serial port. It shows that the seismic data acquisition system we designed can achieve the functions of seismic signal acquisition, digitization, and transmission.

## 4. Discussion

This paper has completed the design and experimental research of a seismic data acquisition system based on wireless transmission. The system successfully captured seismic data from the PE transducer and transmitted it wirelessly to the gateway unit in real-time.

This paper proposes a PE transducer with a simply supported beam structure using PMN-PT as a piezoelectric element, which has high sensitivity, good low-frequency response performance, and is small in size. The simulation calculation and experimental study were carried out. The results showed that applying PMN-PT to the piezoelectric simply supported beam can increase the model sensitivity by more than 190%, compared to using PZT materials. The sensitivity of the simply supported beam model is higher than that of the commonly used central compressed structure. The fidelity of the simply supported beam is better than that of the central compressed structure under the vibration of low frequencies, indicating that a seismic piezoelectric accelerometer based on the piezoelectric simply supported beam has a wider frequency band and a higher sensitivity.

The piezoelectric transducer converts the vibration signal into an electrical charge signal, and the charge conversion circuit converts it into the voltage signal. After the voltage signal is filtered and amplified, it is collected by an ADC and FPGA to digitize the acceleration signal, which achieves the functions of data acquisition and data transmission of the seismic data acquisition system. The wireless transmission network sends the FPGA data to the monitoring centre to analyse and process the signal for real-time monitoring and measurement.

Further works should be focused on completing a three-component digital accelerometer and a hybrid wireless transmission network based on ZigBee and WiFi.

## 5. Patents

Two Chinese utility model patents have resulted from the work reported in this manuscript.

## Figures and Tables

**Figure 1 sensors-21-04308-f001:**
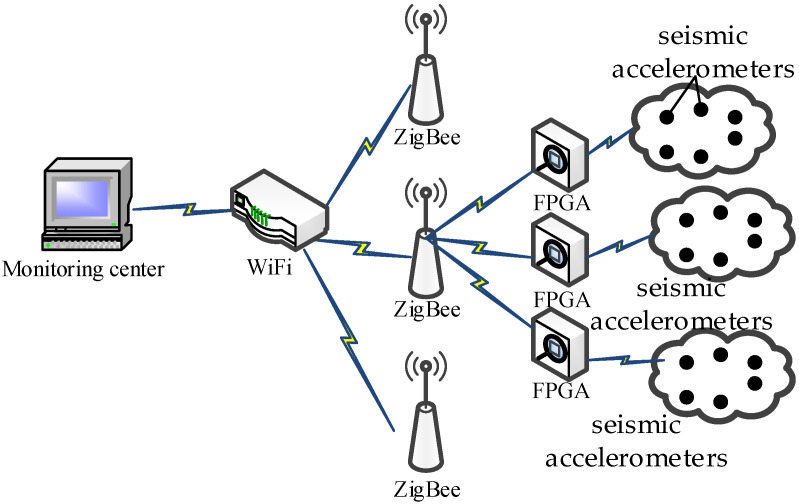
A schematic diagram of seismic data acquisition system, based on wireless transmission.

**Figure 2 sensors-21-04308-f002:**
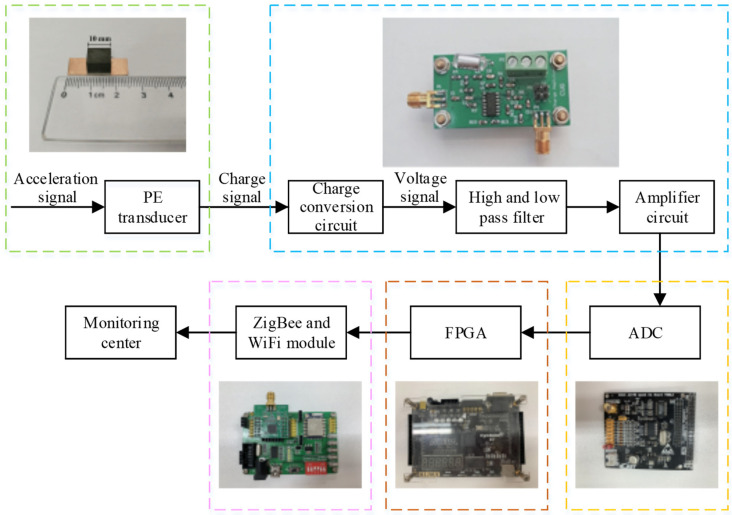
Schematic diagram of seismic data acquisition system based on wireless transmission.

**Figure 3 sensors-21-04308-f003:**
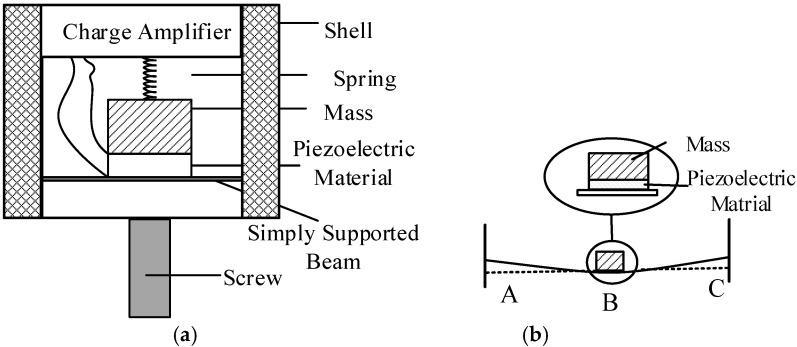
(**a**) The design of the piezoelectric accelerometer; (**b**) The PE transducer based on simply supported beam.

**Figure 4 sensors-21-04308-f004:**
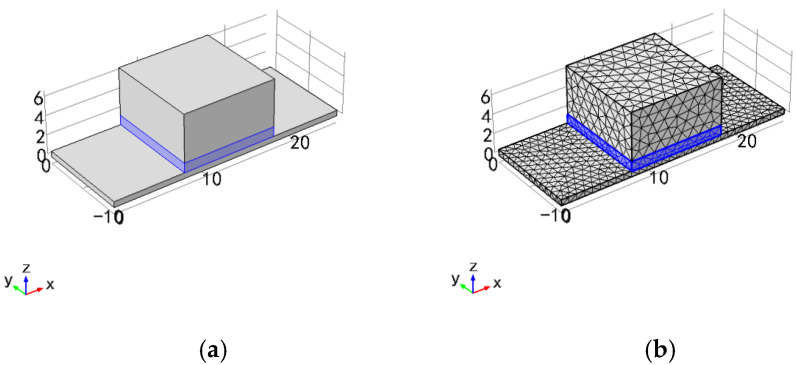
(**a**) The simulation model of the PE transducer; (**b**) Model meshing.

**Figure 5 sensors-21-04308-f005:**
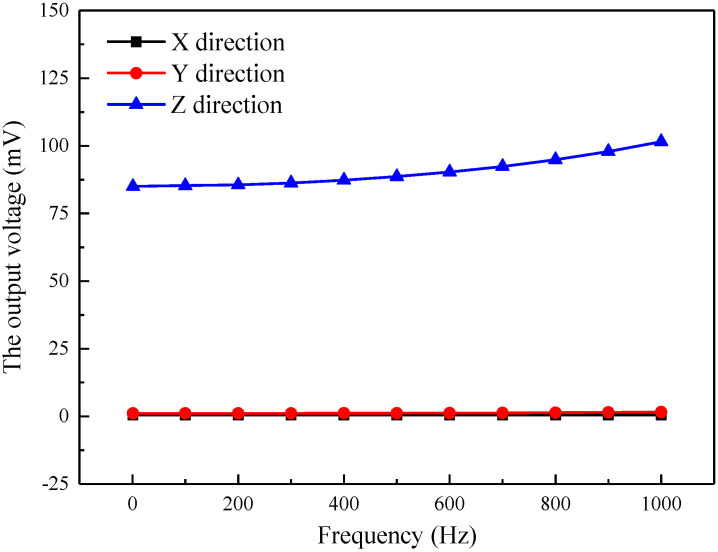
The output voltage curves of the model when vibrated in different directions.

**Figure 6 sensors-21-04308-f006:**
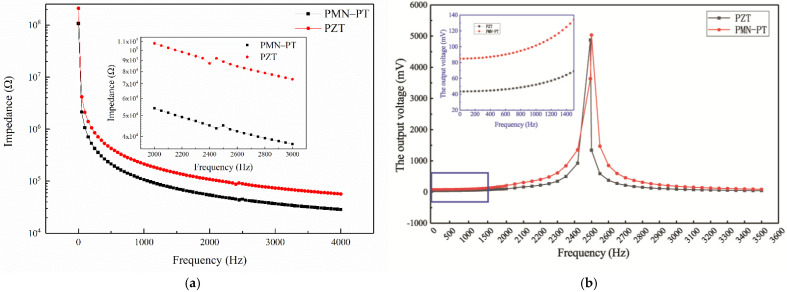
(**a**) The impedance spectrum of PMN-PT and PZT; (**b**) The output voltage of the models based on PMN-PT and PZT.

**Figure 7 sensors-21-04308-f007:**
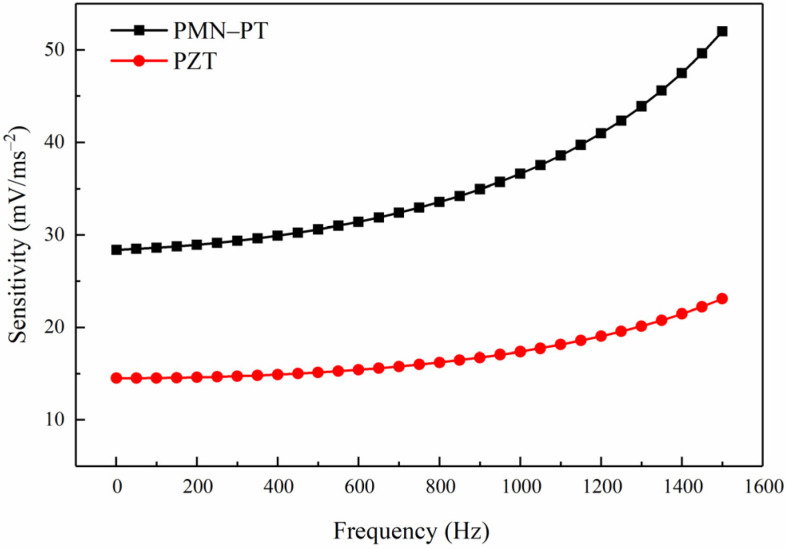
The sensitivity of the models based on PMN-PT and PZT.

**Figure 8 sensors-21-04308-f008:**
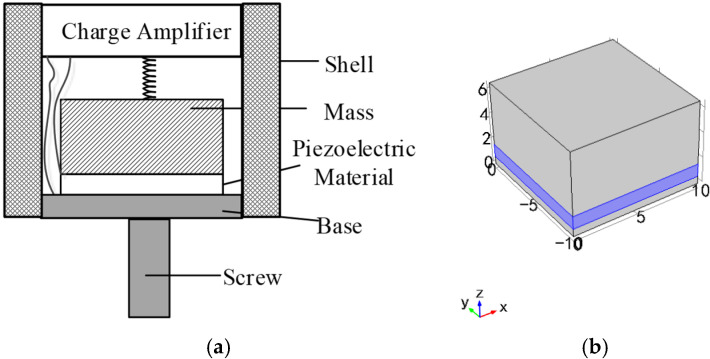
(**a**) The schematic of the central compressed model; (**b**) The central compressed model.

**Figure 9 sensors-21-04308-f009:**
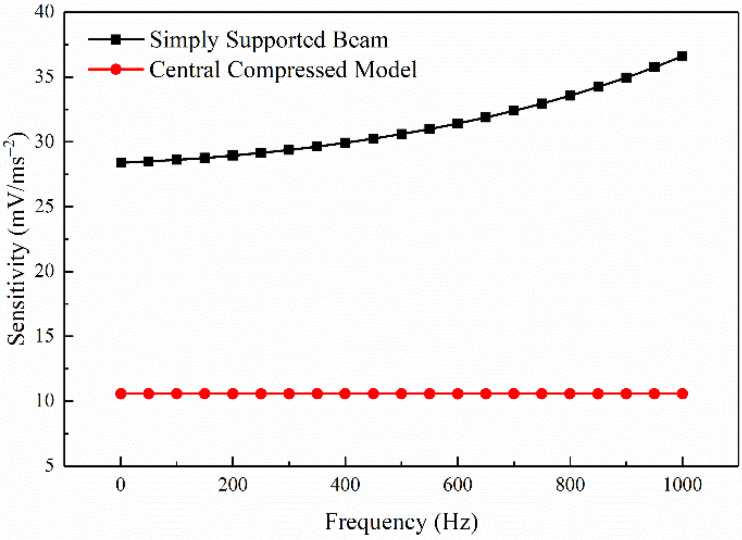
The sensitivity of different structural models.

**Figure 10 sensors-21-04308-f010:**
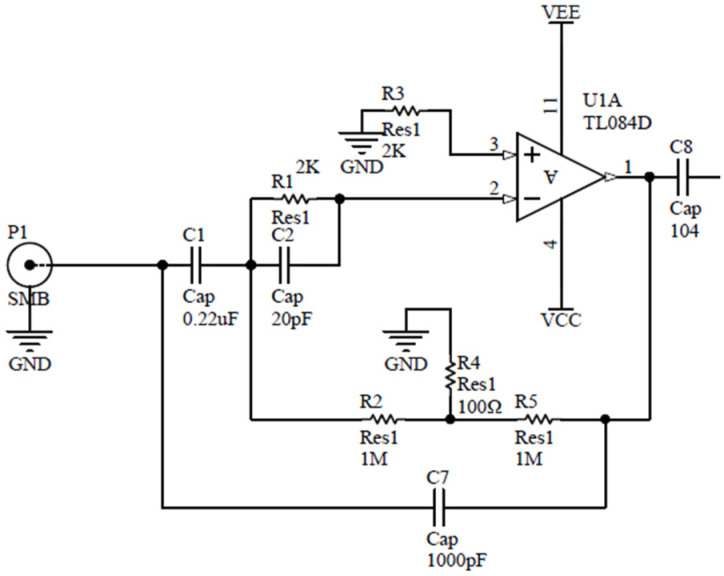
Schematic diagram of charge conversion circuit.

**Figure 11 sensors-21-04308-f011:**
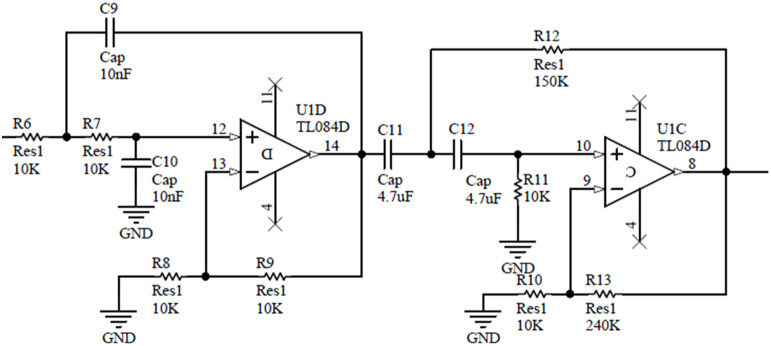
Schematic diagram of low and high-pass filter and amplifier circuit.

**Figure 12 sensors-21-04308-f012:**
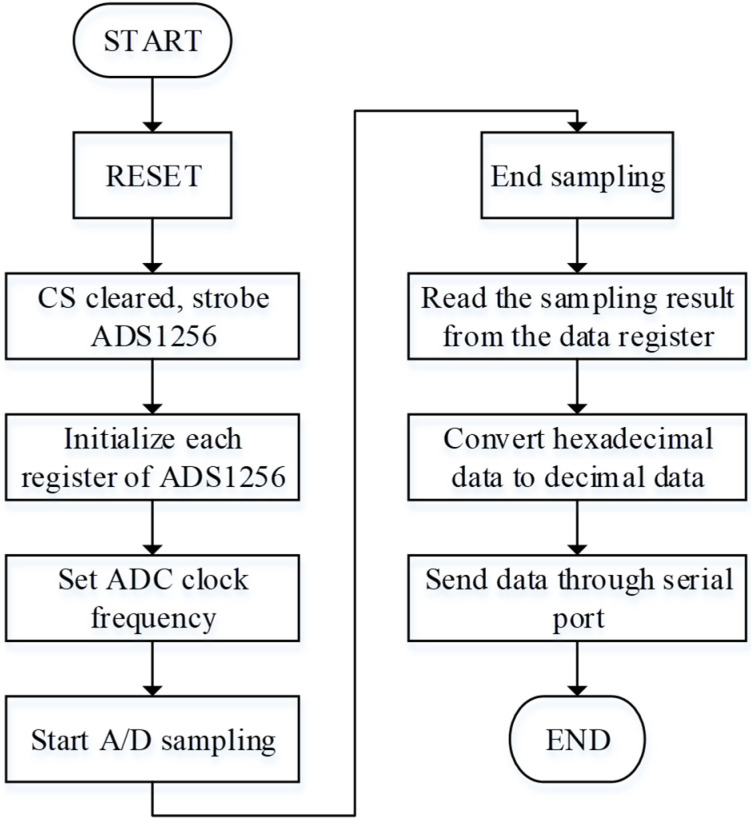
The flowchart of digital data acquisition.

**Figure 13 sensors-21-04308-f013:**
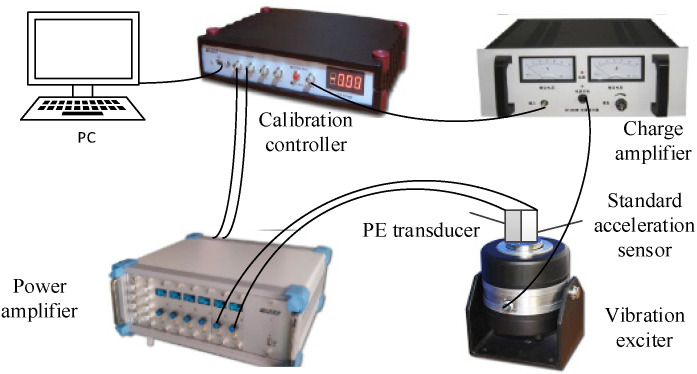
The structure diagram of the control system.

**Figure 14 sensors-21-04308-f014:**
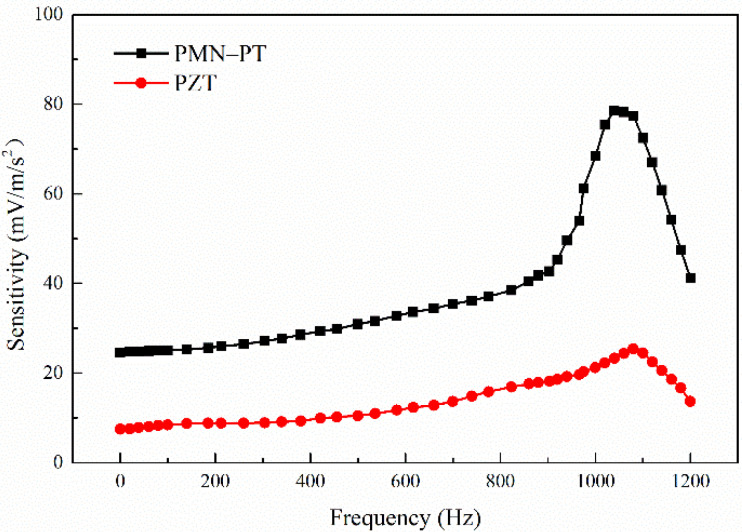
The experimental sensitivity of the simply supported beam transducer is based on PMN-PT and PZT.

**Figure 15 sensors-21-04308-f015:**
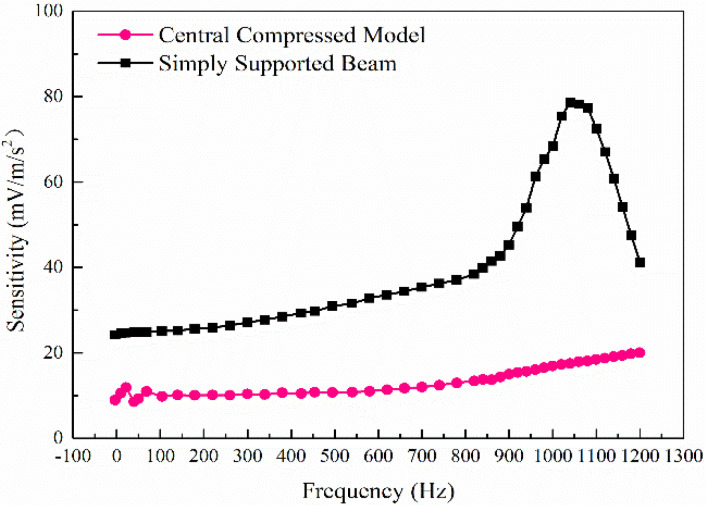
The sensitivity of different structural models.

**Figure 16 sensors-21-04308-f016:**
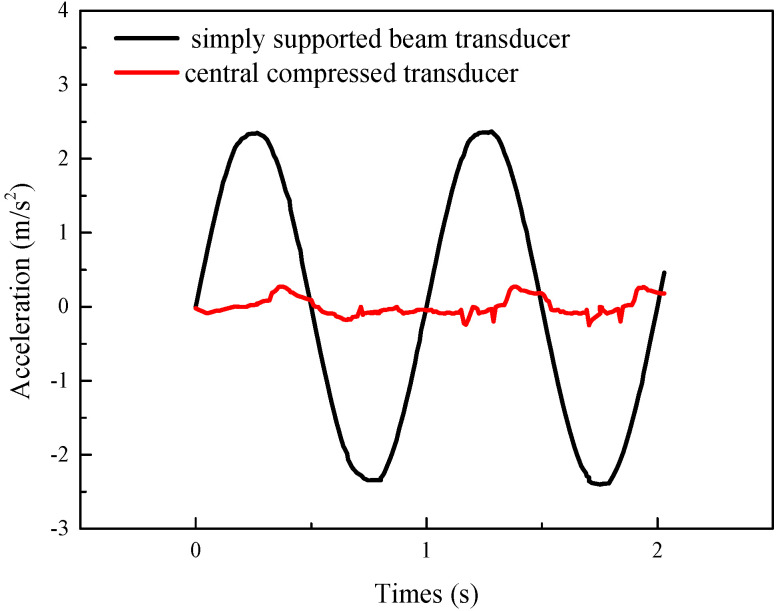
The waveform of different models at 1 Hz.

**Figure 17 sensors-21-04308-f017:**
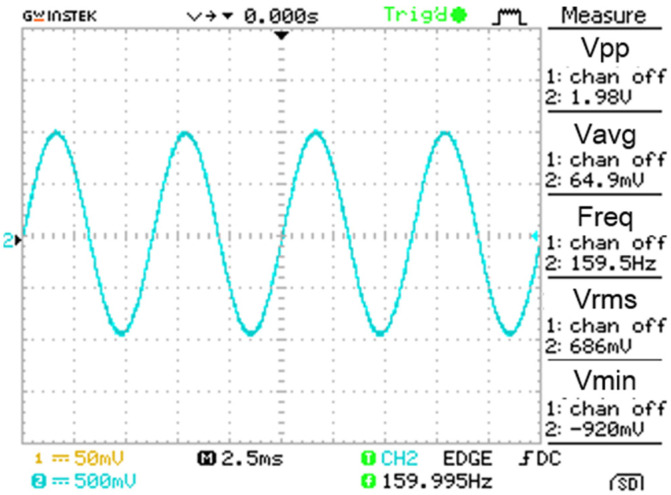
The output of the designed accelerometer with the charge amplifier circuit.

**Figure 18 sensors-21-04308-f018:**
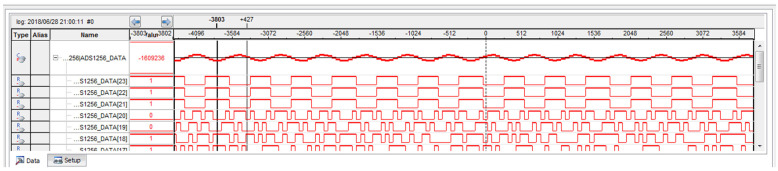
The signal collected by FPGA is displayed in SignalTap.

**Table 1 sensors-21-04308-t001:** Comparison of some element parameters between PMN-PT and PZT-5A.

Material	$\varepsilon_{33}^{T}/\varepsilon_{0}$	Tanδ/%	*d*_33_/(pC/N)	*k_t_*	*k*_33_/%	$s_{33}^{E}$/(pm^2^/N)	Ρ/(kg/m^3^)
PMN-PT	6000	<1.0	2000	0.61	94	54.3	8100
PZT-5A	1300	0.4	289	0.51	70	12.3	7500

**Table 2 sensors-21-04308-t002:** The dimensions of piezoelectric elements and the simply supported beam used in this work.

Composition	Material	Dimension (mm × mm × mm)
Piezoelectric element I	PMN-PT	10 × 10 × 1
Piezoelectric element II	PZT	10 × 10 × 1
Mass	W	10 × 10 × 5
Simply supported beam	BeCu	25 × 10 × 0.6

**Table 3 sensors-21-04308-t003:** The Sensitivity of the transducers of different model.

Transducers	Voltage Sensitivity of Central Compressed Model (Unit mV/ms^−2^)	Voltage Sensitivity of Simply Supported Beam (Unit mV/ms^−2^)	Charge Sensitivity of Central Compressed Model (Unit pC/ms^−2^)	Charge Sensitivity of Simply Supported Beam (Unit pC/ms^−2^)
Based on PMN-PT	18.5	24.8	1.85	2.48
Based on PZT	8.2	8.44	0.82	0.844
Standard sensor	1.29	/	0.129	/

## Data Availability

Not applicable.

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
