# Peer review of "A Seismic Data Acquisition System Based on Wireless Network Transmission"

_sensors, 2021, doi:10.3390/s21134308_

Round 1

Reviewer 1 Report

Report on paper "A seismic data acquisition system based on wireless network transmission"submitted by Huang et al., for publication in Sensors (sensors-1222665).

The authors designed a seismic data acquisition system based on wireless network transmission in order to improve the performances of existing acquisition system. To do so, they proposed a PE transducer with a simply supported beam structure using PMN-PT as piezoelectric element while the charge conversion, filtering, amplification and digitization of the output signal are processed for a wireless transmission to the monitoring center. Although the paper is interesting and contains theoretical and experimental results, it cannot be accepted in its present form and the authors must perform some modifications by addressing the following comments:

  1. In the introduction, the literature survey lacks of references in the field of resonant accelerometers, which is a topic deeply investigated in the recent past.
  2. Section 2 should be extended and full details about the design and FE model (type of elements, mode size, solver, convergence, stability…) should be given.
  3. In figure 5(b), the frequency step size should be decreased in the region close to the resonance peak.
  4. In figure 8, the authors should explain why the sensitivity of the central compressed model is almost constant with respect to the frequency.
  5. In figure 14, what happens between 1000 Hz and 1100 Hz? Why the authors did not perform their simulations in the same frequency domain as the experimental tests of Figure 13?
  6. The author should explain how the damping was implemented in the FE model. Is it the measured damping? How it was measured? Which kind of damping has been used?
  7. What are the reasons explaining the differences between theoretical and experimental results (figures 13 and 14)?
  8. The proposed model, which is linear, is limited by the onset of nonlinearity. The author should discuss that with respect to the recent literature related to resonant sensors operating in the nonlinear regime (for instance [Nonlinear Dynamics, 67, 859–883, 2012], [Appl. Phys. Lett. 117, 033502 (2020)], [Nonlinear Dynamics, 54, 93–122 (2008)]).
  9. Important specifications such as shock resistance and transverse sensitivity could be taken into consideration and discussed.
  10. What about the robustness of the proposed device against temperature variations that may affect the sensor resonance frequencies?
  11. The description of the results lacks of depth and the authors should explain in detail each result according to the physical phenomena and the significance of each result for accelerometer designers.
  12. The quality of some figures should be enhanced.

Reviewer 2 Report

In this paper, a seismic data acquisition system based on wireless network transmission is proposed to improve the low-frequency response and low sensitivity. Extensive experiments are conduct to validate the performance of the proposed system.

Strengths:

  • Data acquisition system is an important topic that worth to study intensively.
  • The proposed system has novelty and can make sense.
  • Writing and organization of the paper are good.

Weaknesses:

  • Figure 2 is not clear enough to see.
  • Some Figures are not centered, such as Figure 1, Figure 3, and Figure 4. 

Round 2

Reviewer 1 Report

The authors have addressed my comments sufficiently to recommend publication of the paper in its current form.